# *O*-GlcNAc transferase plays a non-catalytic role in *C. elegans* male fertility

Daniel Konzman[1,2], Tetsunari Fukushige[3], Mesgana Dagnachew[1], Michael Krause[3], John A. Hanover[1]*

1 Laboratory of Cellular and Molecular Biology, National Institute of Diabetes and Digestive and Kidney Diseases, National Institutes of Health, Bethesda, Maryland, United States of America, 2 Department of Biology, Johns Hopkins University, Baltimore, Maryland, United States of America, 3 Laboratory of Molecular Biology, National Institute of Diabetes and Digestive and Kidney Diseases, National Institutes of Health, Bethesda, Maryland, United States of America

* jah@helix.nih.gov

**Data Availability Statement:** All relevant data are within the manuscript and its Supporting Information files.

**Funding:** This work was made possible by funding from the Intramural Research Program of the

## Abstract

Animal behavior is influenced by the competing drives to maintain energy and to reproduce. The balance between these evolutionary pressures and how nutrient signaling pathways intersect with mating remains unclear. The nutrient sensor *O*-GlcNAc transferase, which post-translationally modifies intracellular proteins with a single monosaccharide, is responsive to cellular nutrient status and regulates diverse biological processes. Though essential in most metazoans, *O*-GlcNAc transferase (*ogt-1*) is dispensable in *Caenorhabditis elegans*, allowing genetic analysis of its physiological roles. Compared to control, *ogt-1* males had a four-fold reduction in mean offspring, with nearly two thirds producing zero progeny. Interestingly, we found that *ogt-1* males transferred sperm less often, and virgin males had reduced sperm count. *ogt-1* males were also less likely to engage in mate-searching and mate-response behaviors. Surprisingly, we found normal fertility for males with hypodermal expression of *ogt-1* and for *ogt-1* strains with catalytic-dead mutations. This suggests OGT-1 serves a non-catalytic function in the hypodermis impacting male fertility and mating behavior. This study builds upon research on the nutrient sensor *O*-GlcNAc transferase and demonstrates a role it plays in the interplay between the evolutionary drives for reproduction and survival.

## Author summary

Animals must make decisions on whether to engage in reproduction or to conserve energy. These decisions must take into account the energy available to the animal, therefore making the nutrient sensing enzyme OGT of particular interest. In response to nutrient levels in the cell, OGT transfers the GlcNAc sugar onto proteins to regulate their function. OGT is implicated in many human diseases including diabetes, cancer, and X-linked intellectual disability. By deleting the gene encoding OGT in the nematode *C. elegans*, we show OGT is required for male fertility. We assessed the behavior of these mutant male worms and found they are less likely to seek mates. Surprisingly, expressing

National Institute of Diabetes and Digestive and Kidney Diseases, National Institutes of Health, intramural research program grant ZIADK060103 to JAH. The funders had no role in study design, data collection and analysis, decision to publish, or preparation of the manuscript.

**Competing interests:** The authors have declared that no competing interests exist.

OGT specifically in the hypodermis was able to raise male fertility and restore behavior to normal levels. In addition, mutations which prevent OGT from transferring GlcNAc do not negatively impact fertility, suggesting a different function of OGT is necessary in this process. Our study demonstrates that OGT is important in critical behavioral decisions and that further investigation in *C. elegans* may help reveal new functions of the enzyme.

## Introduction

The survival of an organism depends on its ability to respond to the environment around it. To maintain life and health, organisms must adequately respond to changes in factors such as temperature, toxins, and nutrition. Critically, reproduction is impacted by many environmental factors including diet, xenobiotics, and stress [1]. Two key evolutionary pressures, survival and reproduction, are both impacted by the environment and can conflict with each other. Survival is supported by the conservation of energy, acquisition of nutrients, and the avoidance of risk. Conversely, reproduction entails risks and requires extensive energy use through gametogenesis, mating, and the raising of young. The behavioral prioritization between these two drives is complex and is particularly impacted by nutrient status [2–4].

With its quick generation time and genetic amenability, the nematode *Caenorhabditis elegans* (*C. elegans*) provides a good model to investigate reproduction and behavior. While the self-fertile hermaphrodites have no need to mate in order to reproduce, males must seek out mates and go through a complex series of behaviors to successfully sire progeny [5]. After successfully locating a mate, the male must make contact with its tail, start backwards locomotion, locate the vulva, and transfer sperm. If the male starts on the side without the vulva or doesn't detect it at first pass, it will need to execute a tight turn of its tail to switch sides and continue searching. Mate-seeking behavior of males is impacted by nutrition and related pathways such as the insulin-like signaling pathway [3], making the nutrient-sensing enzyme *O*-GlcNAc transferase (OGT) a potential regulator of mating.

OGT transfers GlcNAc to serine and threonine residues of its protein targets to regulate many processes within cells. To transfer GlcNAc, OGT requires UDP-GlcNAc, a nucleotide sugar which varies in concentration depending on cellular nutrient status [6,7]. This nutrient-sensitive post-translational modification is added to thousands of nucleocytoplasmic proteins, and can be dynamically removed by the *O*-GlcNAcase (OGA) [8,9]. *O*-GlcNAc modification of proteins can modify their activity, subcellular localization, and interactions with other proteins [9,10]. *O*-GlcNAc has been shown to regulate a diverse array of cellular processes including signaling, stress response, and gene expression [6,7,11,12]. OGT has functions other than transferring *O*-GlcNAc including HCF-1 cleavage [13], and non-catalytic roles such as protein-protein interactions, which are all critical for growth of mammalian cells [14]. However, the specific cellular pathways and physiological process reliant on non-catalytic functions of OGT remain largely unknown. Deregulation of *O*-GlcNAcylation is a feature of human disorders including diabetes, cancer, and Alzheimer's disease [10,15], in addition to the congenital disorder associated with OGT mutations: OGT-X-linked intellectual disability (XLID) [16,17].

Though OGT is essential in most metazoans [18], loss of *O*-GlcNAc transferase (*ogt-1*) in *C. elegans* is viable [19]. As the only model organism that tolerates loss of OGT, *C. elegans* is uniquely suited for genetic analysis of the biological roles of this enzyme. *ogt-1* mutant worms have proven to be a powerful model of insulin resistance [7,19,20], stress [21–25], neurological disorders [18,26–29], and for identifying non-catalytic roles of OGT [25,30,31]. Highlighting its importance in metabolic regulation, *ogt-1* deletion causes altered macronutrient storage,

including a three-fold reduction in lipid stores and three-fold increase in glycogen and treha-lose stores [19]. These animals also have altered entry into diapause states when starved, being more likely to form dauers [19,32], and less likely to enter adult reproductive diapause [31]. Developmental changes have also been noted, including a decreased lifespan [22,33] and increased susceptibility to transdifferentiation [34,35]. Further, *ogt-1* mutant worms have been used to uncover the role of OGT in such behaviors as hypersensitivity to touch and altered habituation to repeated stimuli [36]. While *ogt-1* hermaphrodites have only a slight reduction in brood size in normal husbandry conditions [21,24,33,37], we noticed males produce very few offspring in crosses, despite no obvious uncoordinated movement phenotypes. Here we detail the *ogt-1* male fertility defect and provide evidence this phenotype relies on a non-cata-lytic role of OGT-1 in the hypodermis that results in behavioral prioritization away from mate-seeking behavior.

## Results

### *ogt-1* is required for male fertility

We set out to determine the impact that *ogt-1* deletion has on male fertility. This was assessed by mating them with feminized worms and counting their progeny. *fem-1(hc17)* worms raised at 25˚C produce no sperm and thus have no self-progeny, but produce offspring after mating with a fertile male [38]. In this experiment, the two control lines (N2 and *him-5*) have similarly high progeny counts (Fig 1A), and thus *him-5* was used as the genetic background "wild-type" throughout this study as the "high incidence of males" phenotype (Him) provides adequate numbers of males to conduct experiments [39]. We tested multiple alleles of *ogt-1* in the *him-5* background for male fertility and compared them with controls. *fem-1* worms mated to males with deletions in *ogt-1* had severely reduced 24-hour broods compared with both *him-5* and N2 males. Three independent *ogt-1* alleles displayed this male fertility defect: our CRISPR dele-tion of the entire *ogt-1* coding region (*jah01*) [31], and two smaller deletions within the coding sequence both expected to produce no functional protein (*ok430* [19] and *ok1474* [21]) (Fig 1A). As the phenotype was strongest in the CRISPR deletion, we used this line throughout the study and refer to the allele as simply "*ogt-1*" unless noted otherwise. Previous studies have shown reduced brood sizes from *ogt-1* deletion hermaphrodites, with a decrease from wild-type that ranges from 10% to 30% [21,24,31]. Mean *ogt-1* male brood size was 65% to 80% lower than the wild-type controls (Fig 1A). While the progeny count from *ogt-1* males was often very low or zero, all three alleles had a wide range of values (Fig 1A). We also assessed the fertility of males harboring mutations in *oga-1*, the gene encoding the enzyme that removes *O*-GlcNAc from proteins. Two *oga-1* alleles (CRISPR deletion of the entire *oga-1* coding region *av82* [31], and *ok1207* [20]) did not appear to affect male fertility, as their progeny counts were similar to positive controls (Fig 1A). Analysis of the *ogt-1(jah01);oga-1(av82)* double CRISPR deletion line showed low fertility similar to *ogt-1(jah01)* alone (Fig 1A).

Due to the ubiquitous expression of *ogt-1* throughout development [20,25,34,40] and the complexity of reproductive biology, the deficit in male fertility phenotype may originate from many different tissues. To determine the tissue in which *ogt-1* expression is critical for normal fertility, we performed a series of tissue-specific rescue experiments. For these experiments, we used the CRISPR deletion of the whole gene, *ogt-1(jah01)*, as the background for genetic res-cue. Through microinjection, we first established two independent lines carrying a fosmid containing the *ogt-1* gene as an extrachromosomal array. Males with the fosmid had wild-type fertility when mated with *fem-1* worms (Fig 1B), demonstrating a robust rescue of the pheno-type. To determine if functional OGT-1 was being produced by the fosmid-rescued lines, global *O*-GlcNAcylation was assessed by western blot. The blot showed *O*-GlcNAc levels were

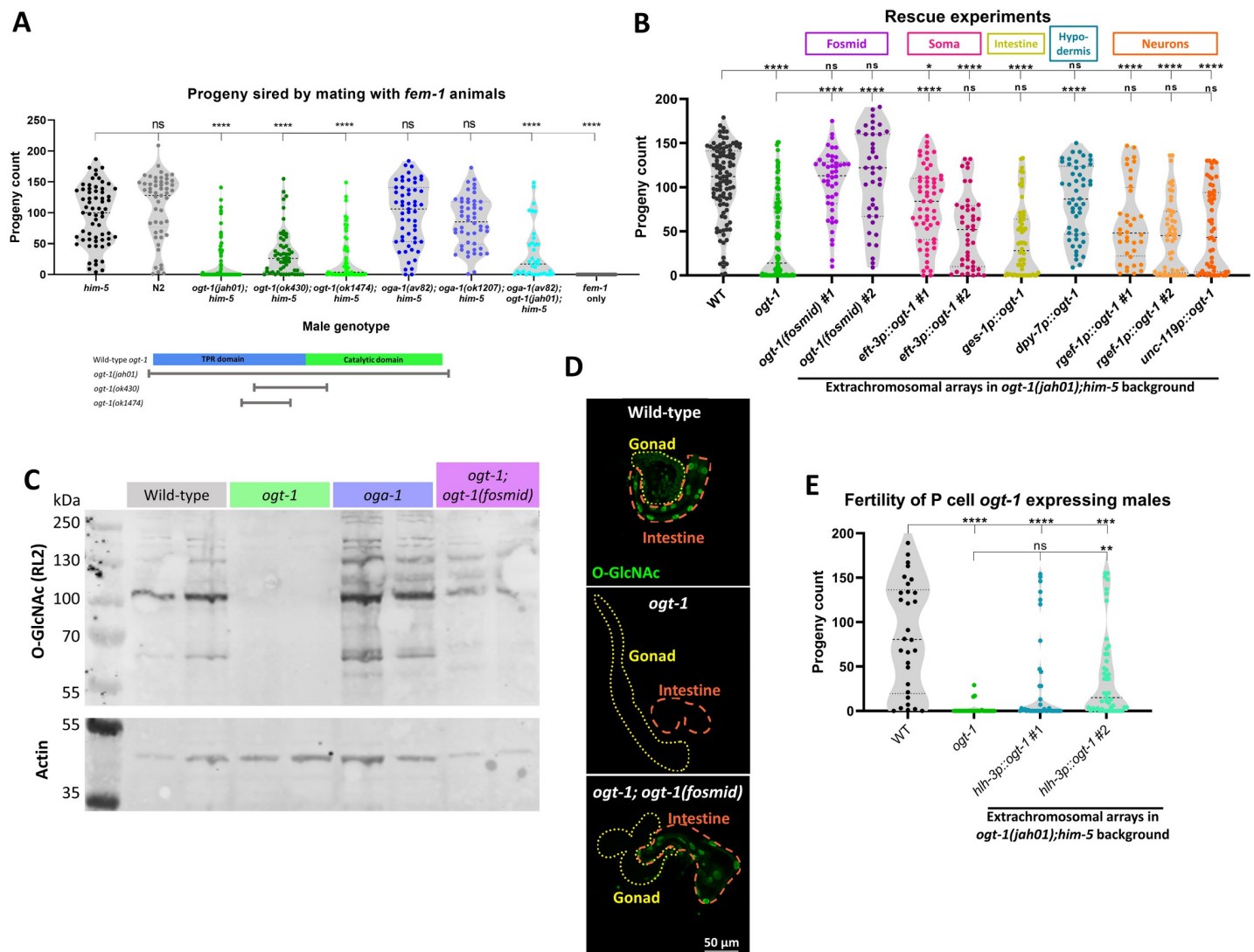

**Fig 1. *ogt-1* mutations reduce male fertility and can be rescued by expression in the hypodermis.** (A) 24-hour brood count from self-sterile *fem-1* worms mated with males of each specified genotype, shown as individual values overlaid on violin plots with quartiles. Statistical comparisons to *him-5* by one-way ANOVA are shown above. Below, a schematic of the three *ogt-1* deletion alleles in the context of the OGT-1 protein structure, with the tetracopeptide repeat (TPR) domain in blue and the catalytic domain in bright green. (B) 24-hour brood by various *ogt-1* rescue males mating with *fem-1*. All extrachromosomal array rescue lines are in the deletion background *ogt-1(jah01)*. Statistical comparisons by one-way ANOVA to WT and *ogt-1(jah01)* are shown above. (C) Western blot of worm lysates demonstrates rescue of anti-*O*-GlcNAc antibody (RL2) signal in fosmid-rescued line, with anti-actin antibody signal shown for comparison. (D) Representative immunohistochemistry images of dissected males shows *O*-GlcNAc staining is restored in somatic tissues (intestine, dashed orange line) of the fosmid-rescued line, but not in the germline (gonad, dotted yellow line). (E) 24-hour brood count of *hlh-3p::ogt-1* rescue lines by mating with *fem-1* with statistical comparisons to WT and *ogt-1* by one-way ANOVA shown above. All males in panels B-E are in the *him-5* background. * = p<0.05, ** = p<0.01, *** = p<0.001, **** = p<0.0001, ns = not significant.

undetectable in *ogt-1(jah01)* worms (Fig 1C, lanes 3–4), and that the fosmid-rescued line restored global *O*-GlcNAcylation to wild-type levels (Fig 1C, lanes 7–8), but not as high as the elevated levels detected from *oga-1(av82)* worms (Fig 1C, lanes 5–6). As an extrachromosomal array, expression of the *ogt-1* transgene was likely repressed in the germline [41], and we used antibody staining on the gonad and intestine of dissected males to test this. *O*-GlcNAc staining was present in both the intestine and gonad of wild-type worms, absent from both tissues in *ogt-1(jah01)* worms, and present in the intestine but not the gonad of the fosmid-carrying worms (Fig 1D). This suggests the fosmid enabled *ogt-1* expression in somatic tissues but not

in the germline. With the fertility rescue, this indicates somatic expression of *ogt-1* was sufficient for normal fertility.

Tissue-specific rescue plasmids were constructed using promoter fragments inserted directly upstream of the *ogt-1* cDNA containing two endogenous introns (see Materials and Methods for more detail). The same promoters were also cloned into a GFP plasmid and co-injected to ensure the promoters were expressing as expected. Driving expression of *ogt-1* by the pan-somatic *eft-3* promoter [42] was sufficient to rescue male fertility (Fig 1B), which confirmed the fosmid rescue results and that the *ogt-1* transgene plasmid was functional. Expression of *ogt-1* from the hypodermal promoter *dpy-7* [43] was also sufficient to rescue fertility (Fig 1B), while the intestinal promoter *ges-1* [43] and neuronal promoters *rgef-1* [44] and *unc-119* [45] did not rescue fertility (Fig 1B).

The *dpy-7* promoter is frequently used to drive hypodermal expression in tissue-specific rescue experiments [46], but careful analysis of *dpy-7* expression has shown it turns on in embryonic and post-embryonic P cells [47]. Through development, the P cells produce many daughter cells which differentiate into both hypodermal cells and neurons [48]. Of particular interest are P10 and P11, which in the male produce several cells that form the hook sensillum and other male tail neurons [48]. This suggests the possibility that in the *dpy-7p::ogt-1* rescue line, early expression of *ogt-1* in P10 and P11 could be inherited by neuronal daughter cells which ultimately play a critical role in the adult male tail. We tested this idea genetically with an additional rescue experiment using the *hlh-3* promoter, which expresses in the early P cells like the *dpy-7* promoter, but not in hypodermal cells later in development [49]. Two independent *hlh-3p::ogt-1* lines showed fertility significantly lower than wild-type (Fig 1E), indicating a failure of P cell expression to rescue the *ogt-1* fertility phenotype. With the finding that the pan-neuronal promoters of *rgef-1* and *unc-119* also did not rescue fertility (Fig 1C), this suggests it is unlikely that *ogt-1* contributes to male fertility through its expression in P cell descendent neurons. These data support the interpretation that the *dpy-7p::ogt-1* construct rescues fertility due to its expression in the hypodermis, not its expression in the early P cells.

Considering the robust fertility rescue by the hypodermal expression, male tail development was an appealing hypothesis to explain the mating defect. Some structures key to the mating process, such as the sensory rays, are derived from hypodermal cells [48,50]. Analysis of high-magnification images taken of male tails revealed either a slight increase in developmental defects or no change from wild-type, depending upon allele (S3 Fig). The full-gene CRISPR deletion *ogt-1(jah01)* and smaller deletion *ogt-1(ok430)* both showed no significant elevation in defects. Twelve percent of *ogt-1(ok1474)* had developmental defects including missing rays or fusions of multiple rays, while these defects occurred in only 2% of wild-type males (S3 Fig). The defects we observed were primarily within the V6 rays (rays 4–6). Because this phenotype is specific to one allele and relatively mild (88% of males appear normal), it is unlikely to be a major contributor to *ogt-1* male infertility.

## *ogt-1* males transfer sperm less often and have reduced sperm count

We next performed mating experiments to examine the cause of the fertility defect in greater detail. When a *C. elegans* male mates with a self-fertile hermaphrodite, the male sperm typically outcompete the hermaphrodite sperm, at nearly 100% efficiency [51]. To test if *ogt-1* male sperm can compete with hermaphrodite sperm, males were crossed with self-fertile *unc-39(ct74)* hermaphrodites. Self-progeny from *unc-39(ct74)* hermaphrodites had the uncoordinated (Unc) phenotype, while F1s resulting from male sperm were non-Unc. As expected, mating with wild-type males resulted in near 100% outcross progeny (Fig 2A), indicating successful sperm transfer and that the sperm was highly competitive. *ogt-1* males showed a

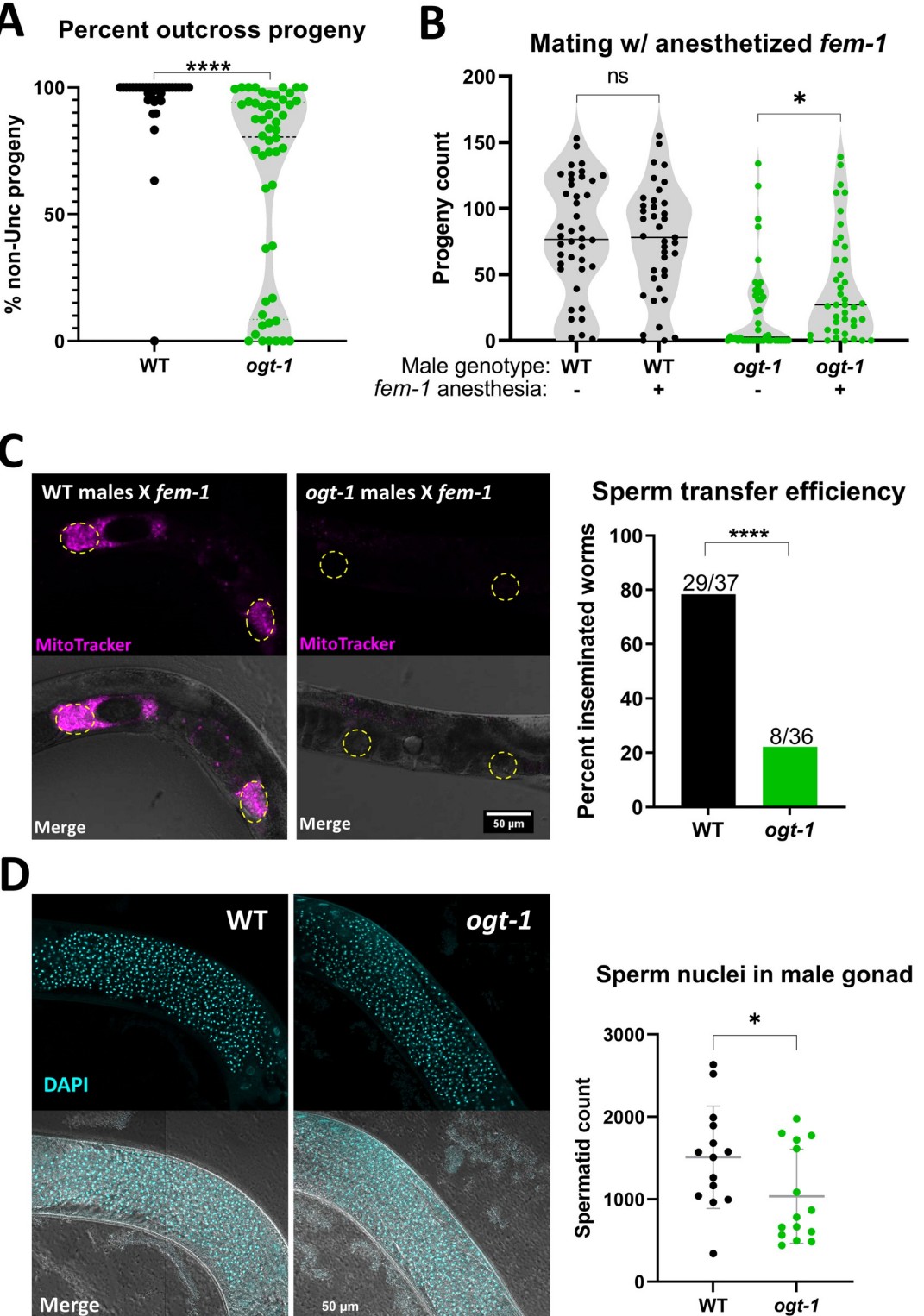

**Fig 2. *ogt-1* males transfer sperm less often and have reduced sperm count.** (A) Sperm competition tested by mating males with *unc-39* hermaphrodites shows lower percent outcross progeny for *ogt-1(jah01)* males, as assessed by t-test. (B) *ogt-1 (jah01)* males show higher progeny counts when mating with anesthetized *fem-1* worms, as assessed by t-test. (C) Representative images of unlabeled, unanesthetized *fem-1* animals after mating for one hour with MitoTracker-labeled males (magenta) of indicated genotype. Spermathecae are indicated with dashed yellow circles. Number of successful (any sperm in

reproductive tract) versus unsuccessful sperm transfers (no detectable sperm) was compared between genotypes by Fisher's exact test. (D) Representative images of DAPI-stained (cyan) virgin young adult males, where spermatid nuclei appear as bright, compact puncta. Plot on the right shows sperm count from all images assessed, with mean, standard deviation, and t-test results shown. All males are in the *him-5* background. * = p<0.05, **** = p<0.0001, ns = not significant.

bimodal distribution in percent outcross progeny, with many producing very high proportions of outcross progeny, and others showing low percentages or zero (Fig 2A). While *ogt-1* male sperm may have a competitive disadvantage, differences in mating behavior may also be important, as the slow-moving Unc hermaphrodites are easier to mate with than worms without movement defects.

To directly test if crossing with *ogt-1* males with immobile mates results in more offspring, we tested mating with *fem-1* worms with and without anesthetic. Crossing with wild-type males resulted in many offspring whether or not their *fem-1* mates were anesthetized (Fig 2B). *ogt-1* males produced significantly more offspring when mating with anesthetized worms, but progeny counts were still lower than wild-type (Fig 2B).

Sperm transfer can be directly assessed by tracking fluorescently-labelled male-derived sperm within the reproductive tract of non-labelled mates. Unanesthetized *fem-1* worms were used as mates for consistency with the other experiments and to reduce the likelihood of ovulation which could displace male sperm. After one hour of mating, 78% of *fem-1* worms carried labelled sperm from wild-type males in their spermathecae, while only 22% of those mated with *ogt-1* males did (Fig 2C). Images of the *fem-1* worms following successful sperm transfer showed there was typically less fluorescent signal present in worms mated with *ogt-1* males (S4 Fig), suggesting fewer sperm were transferred. Labelled sperm from either genotype appeared concentrated in the spermatheca (S4 Fig), consistent with *ogt-1* sperm correctly localizing to the spermatheca as would be expected of sperm with normal activation, guidance, and motility. Analysis of high-resolution z-stacks of DAPI stained virgin males revealed the number of spermatids in *ogt-1* males averages just over 1,000, about one third less than the wild-type average near 1,500 (Fig 2D).

### *ogt-1* males exhibit aberrant mating behavior

Due to *ogt-1* males' better performance with slow-moving mates, we hypothesized they may have critical defects in mating behavior. The series of behaviors the adult male *C. elegans* must orchestrate in order to successfully mate are complex but well characterized [5]. A male must first locate a mate, then respond to physical contact by holding the ventral side of its tail against the hermaphrodite. The male will then search for the vulva by moving backwards along the length of the worm. When it reaches either the head or tail of the hermaphrodite, it will perform a tight ventral turn to then search the other side for the vulva. Finally, the male must stop at the vulva, insert its spicules, and ejaculate into its mate.

Before the process of mating can begin, the male must actively seek out mates. The food-leaving assay was developed to measure the mate-searching behavior of *C. elegans* males [3]. A lone adult male placed in a tiny lawn of OP50 will typically exhibit food-leaving behavior, in which they leave the food source and explore extensively, thought to be due to its drive to find a mate (Fig 3A), while hermaphrodites will rarely leave the food source [3,4]. At the endpoint of the 24h assay, 62% of wild-type males had exhibited food-leaving behavior, while only 38% of *ogt-1* males had done so (Fig 3B). To ensure this difference was not due to a movement defect, we analyzed video of *ogt-1* males using the wrMtrck ImageJ package [52] and found the average movement speed of males was equivalent to wild-type (Fig 3C).

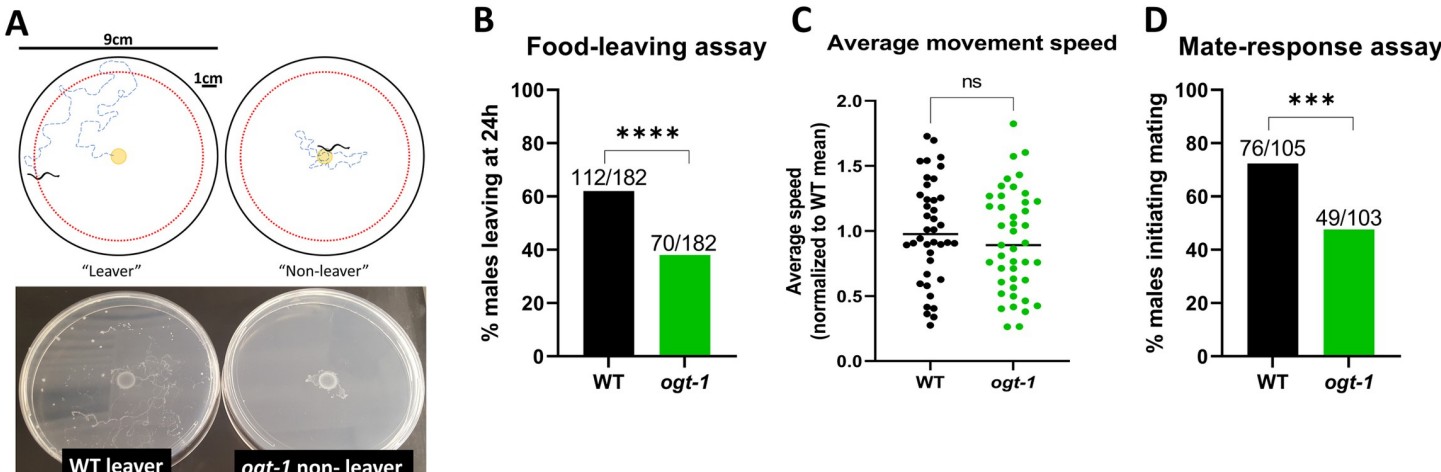

**Fig 3. *ogt-1* males show behavioral differences associated with food-leaving and mate-response.** (A) Schematic of the food-leaving assay: an individual male is placed in a small dot of OP50 (4mm diameter) in the center of a 9cm plate and is scored as a "leaver" if it gets within 1cm of the edge of the plate over the course of the 24h assay. Below, images of representative plates illustrating the exploration of a wild-type leaver and an *ogt-1(jah01)* non-leaver. For this image, plates were kept at room temperature after the assay to allow bacterial growth for better visualization. (B) Food-leaving assay endpoint data (24h) shows *ogt-1(jah01)* males are less likely to exhibit the food-leaving behavior associated with mate-seeking, as assessed by Fisher's exact test. (C) Normalized average moving speed is no different between wild-type and *ogt-1(jah01)* worms, as assessed by t-test. (D) Mate-response assay shows fewer *ogt-1(jah01)* males initiate mating with non-anesthetized *fem-1* animals during a 6-minute assay, as assessed by t-test. All males are in the *him-5* background. *** = p<0.001, **** = p<0.0001, ns = not significant.

As the food-leaving assay assesses male behavior in the absence of mates, we next tested *ogt-1* males' response to the presence of mates. We scored mating initiation with a mate-response assay adapted from prior studies of mating behavior [53,54], in which mating initiation was defined as two seconds of contact between the ventral side of the male tail with its mate. Within the six-minute assay, wild-type males initiated mating 72% of the time, while *ogt-1* males did so only 48% of the time (Fig 3D).

To determine if *ogt-1* deletion caused detriments within further steps of the mating process, videos of males mating with anesthetized *fem-1* worms were analyzed. This analysis revealed that *ogt-1* males exhibit somewhat poorer turn performance while mating (S6A Fig). Using the turn-scoring methodology defined by Loer and Kenyon [55], 12.5% of *ogt-1* males' turns resulted in complete disconnection from their mate ("missed turns"), up from 1.6% in wild-type, and successful turns that involved correction after loss of contact ("sloppy turns") were also more common for *ogt-1* males (7.5%) than wild-type (2.7%) (S6A Fig). Eighty percent of turns performed by *ogt-1* males were scored as good, lower than the 95.7% good turns by wild-type males (S6A Fig).

Though *ogt-1* males successfully located the vulva fewer times overall than their wild-type counterparts (S5 Table), the vulval location efficiency (successful vulval location events divided by total vulval encounters) of *ogt-1* and wild-type were roughly equivalent (S6B Fig). Additionally, wild-type males and *ogt-1* males spent comparable amounts of time at the vulva after stopping (S6C Fig).

## OGT-1 catalytic activity is dispensable for male fertility and mating behavior

We next tested *ogt-1* variants to ascertain which mechanism the OGT-1 protein works through to promote male fertility. Several recent studies have highlighted the importance of non-catalytic functions of OGT [14], including three *C. elegans ogt-1* phenotypes which did not require catalysis [25,30,31]. To determine if OGT-1's catalytic function was necessary for normal male

fertility, we made use of two lines CRISPR-edited with H612A and K957M substitutions, constructed in the *ogt-1(dr84)* (*ogt-1*::*gfp*) strain which has a fluorescent tag (GFP) inserted at the C-terminus of the OGT-1 coding region [25] (Fig 4A). These two sites correspond with the H498A and K842M sites in the human enzyme, which were among the lowest-activity OGT variants tested *in vitro* [56]. Western blot showed undetectable *O*-GlcNAc levels in both the H612A and K957M lines, comparable with *ogt-1* deletion (Fig 4B, lanes 6–7). Putting these alleles in the background of the *oga-1* CRISPR deletion *av82*, which cannot remove *O*-GlcNAc, we determined that H612A had very low levels of activity, as one light band was visible (Fig 4B, lane 8). *ogt-1(K957M);oga-1(av82)* did not show detectable signal (Fig 4B, lane 9), suggesting the K957M mutation more completely abrogates catalysis.

Crossing with males with these missense mutants produced progeny counts comparable with the positive control *ogt-1*::*gfp*, and significantly higher than *ogt-1* deletion (Fig 4C). This demonstrates that male fertility requires OGT-1, but does not require *O*-GlcNAcylation, suggesting the mechanism underlying male fertility is a non-catalytic function of the OGT-1 protein.

As fertility can be impacted in a variety of different ways, we tested the catalytic-dead *ogt-1 (K957M)* line and the hypodermal rescue line in the food-leaving assay to determine if their effects on progeny count were distinct from, or correlated with, mate-seeking. *ogt-1* males had the lowest rate of food-leaving behavior throughout the 24-hour assay, while both *dpy-7p*::*ogt-1* and *ogt-1(K957M)* males were similar to wild-type (Fig 4D). Curve comparison found significant differences between *ogt-1* and each of the other three tested lines. To further explore the behavioral phenotypes of these lines, they were tested with the mate-response assay, which showed *ogt-1* had the lowest percentage of males which responded to mates, while wild-type, *ogt-1(K957M)*, and *dpy-7p*::*ogt-1* had comparable proportions of worms which responded (S8A Fig). Assessment of turn quality again demonstrated *ogt-1* males had the greatest proportion of missed and sloppy turns (S8B Fig). Unlike food-leaving and mate-response, the turning behavior results for *dpy-7p*::*ogt-1* were closer to *ogt-1* than wild-type (S8B Fig), suggesting this behavioral defect isn't rescued by hypodermal expression of *ogt-1*. Wild-type and *ogt-1 (K957M)* were roughly equivalent with very low proportions of missed or sloppy turns (S8B Fig), demonstrating a non-catalytic mechanism is at play in each of these behaviors, as was the case with the overall progeny count (Fig 4C).

The food-leaving (Fig 4D) and mate-response (S8A Fig) results parallel the earlier findings that *dpy-7p*::*ogt-1* and *ogt-1(K957M)* males have wild-type levels of progeny when mating with *fem-1* mates (Figs 1B and 4C). This correlation between behavior and progeny count phenotypes lends support to the hypothesis that the reduced progeny count of *ogt-1* males was caused by behavioral differences.

## Discussion

Here we have described a phenotype of *ogt-1* deletion that negatively impacts the fertility of *C. elegans* males. Three independent alleles of *ogt-1* shared this phenotype, and expression of an *ogt-1* transgene either pan-somatically or specifically in the hypodermis was sufficient to rescue. *ogt-1* deletion males had aberrant mating behavior and reduced sperm count. Experiments with *ogt-1* point mutants revealed that fertility and mating behavior do not depend on OGT-1 catalytic activity.

The finding that expression of *ogt-1* by the *dpy-7* promoter rescued fertility (Fig 1B) suggested the hypodermis is of great importance to this phenotype. The failure to rescue fertility by *ogt-1* expression in the P cells (Fig 1E) confirmed that *ogt-1* is required in the hypodermis for male fertility, and the role it plays is likely later in development. Previously, *ogt-1* was

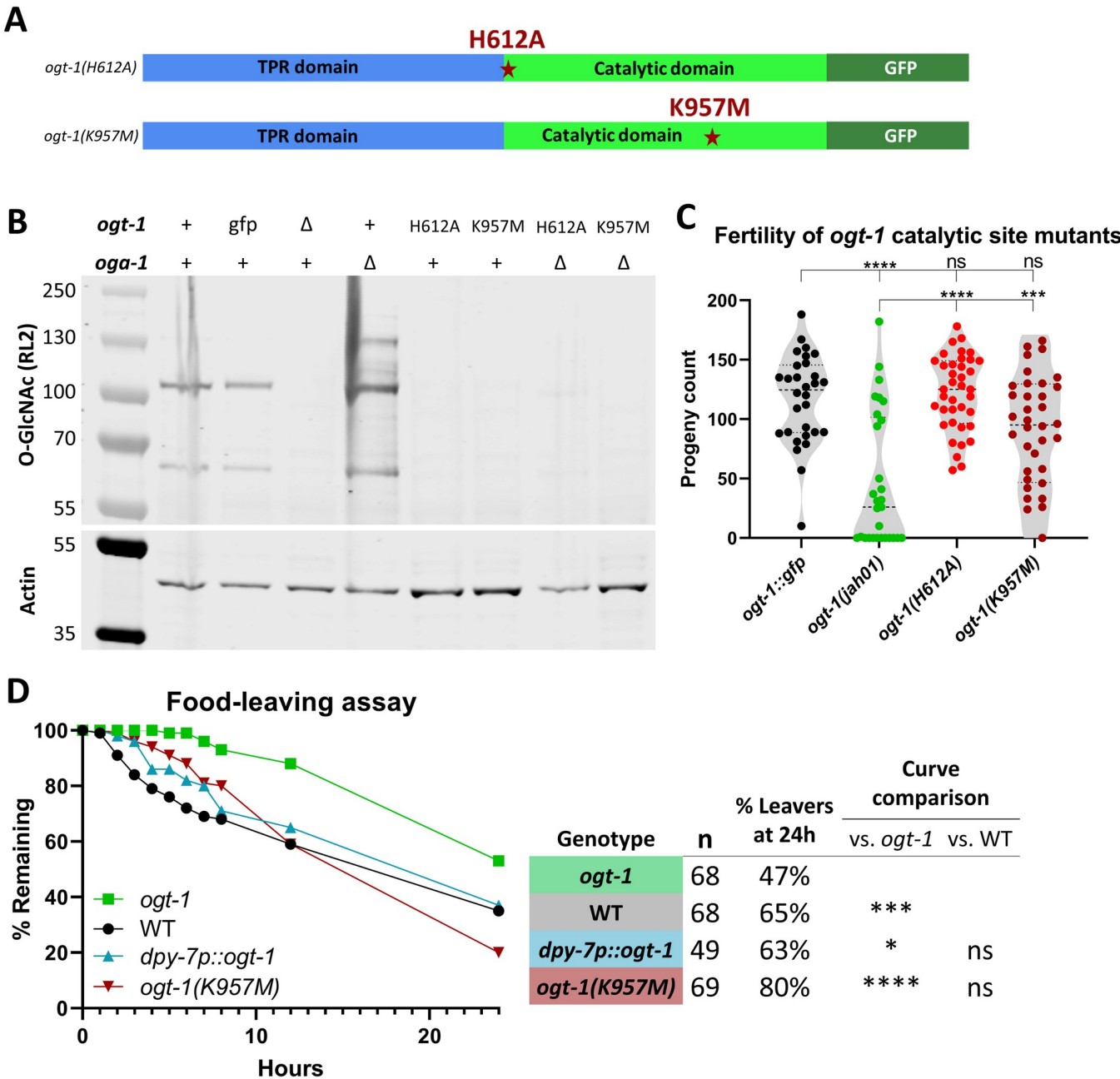

**Fig 4. OGT-1 catalytic activity is not required for male fertility.** (A) Schematic of OGT-1 protein showing the location of two substitutions (dark red stars) within the context of the tetratricopeptide repeat (TPR) domain (blue), catalytic domain (bright green), and GFP tag (dark green). (B) Western blot of worm lysates shows undetectable *O*-GlcNAc signal (RL2 antibody) in *ogt-1Δ (jah01)* and point mutant lines (H612A and K957M). Point mutants are also shown in the background of *oga-1Δ (av82)* background to prevent *O*-GlcNAc removal, thereby increasing any *O*-GlcNAc levels. "gfp" is *ogt-1* with C-terminal GFP sequence (*ogt-1(dr84)*), with similar *O*-GlcNAc signal to WT. Anti-actin antibody signal is shown below for comparison. (C) When mating with *fem-1*, *ogt-1* point mutant males have wild-type level of progeny, as assessed with one-way ANOVA comparisons to *ogt-1::gfp* and *ogt-1(jah01)*. (D) Food-leaving assay time course with datapoints at 0-8h, 12h, and 24h shows *ogt-1(K957M)* and *dpy-7p::ogt-1* males have similar food-leaving behavior wild-type. Includes data from three replications of the experiment. Summary chart showing total males tested (n), the percent of leavers at the endpoint, and statistical comparison of curves by log-rank Mantel-Cox. All males are in the *him-5* background, and *dpy-7p::ogt-1* is also in the *ogt-1(jah01)* background. * = p<0.05, *** = p<0. 001, **** = p<0.0001, ns = not significant.

shown to be required in the hypodermis for the adaptation of worms to hypertonic environmental stress [25]. A screen for genes involved in preventing transdifferentiation identified

*ogt-1* as a gene that helps maintain the hypodermal cell fate [34], a role it also plays in other tissues [35], though whether this has bearing on the mutant phenotypes is unclear. The *C. elegans* hypodermis, beyond its role producing the cuticle in conjunction with each molt [57], also serves important metabolic functions including growth regulation [46,58] and nutrient storage [59]. A recent study compared the gene expression profiles of adult *C. elegans* tissues with human tissues and interestingly found the *C. elegans* hypodermis to be most similar to the human liver, a correlation driven by the hypodermal expression of metabolic enzymes and regulators [60]. The metabolic functions of the hypodermis have previously been shown to play key roles in male mating by storing and mobilizing energy to fuel neurons and muscles directly involved in mating [61,62]. Highlighting the connection between the hypodermis and behavioral prioritization, the hypodermal rescue line showed comparable food-leaving behavior to wild-type (Fig 1B) in addition to rescuing the brood size.

The mating behavior differences between *ogt-1* and wild-type likely play a key role the fertility phenotype. When isolated from mates, adult *C. elegans* males will spontaneously switch their behavioral paradigm from foraging to mate-seeking, which has been studied by observing the rate of food-leaving over 24 hours [3,4]. When tested in this way, *ogt-1* males were less likely to leave the food source, which suggests a reduced sex drive as compared with wild-type males (Fig 3B). When mates are present, male food-leaving behavior is expectedly different, with males no longer engaging in the extensive search for a potential mate [3]. Using an assay with an excess of mates present, we found that *ogt-1* males were less likely to initiate mating (Fig 3D). Lowering the barrier of entry to mating by using anesthetized or slow-moving mates increased the number of progeny produced by *ogt-1* males (Fig 2A and 2B), consistent with a lowered but not completely ablated sex drive. Later in the mating process we observed mild differences in turning behavior between *ogt-1* males and wild-type, while other behaviors such as locating the vulva and dwell time at the vulva remained the same (S6 Fig). While there are likely many contributing factors, the behavioral differences driving this phenotype are likely those before the mating process begins, suggesting *ogt-1* males prioritize mating less than wild-type males.

A model of the conflicting drives between mate-seeking and feeding [4] would predict that since *ogt-1* males were seeking mating less (Fig 3B), they may be prioritizing feeding. This is in line with the role of OGT as a nutrient sensor. Findings in mouse models indicate an evolutionarily-conserved role in regulating feeding behavior, as deletion of OGT in either the PVN neurons [63] or pancreatic α cells [64] induces overeating to the point of obesity. The metabolic profile of *ogt-1* deletion worms is known to be altered, with a 3-fold reduction in lipid stores and 3-fold elevation in glycogen and trehalose stores [19]. Considering prior studies which have indicated a connection between nutrient status and behavioral prioritization, abnormal macronutrient storage or nutrient signaling of *ogt-1* mutants may shift their behavior towards food seeking, resulting in less engagement in mating. Similar to the *ogt-1* food-leaving phenotype, males starved overnight were less likely to leave food in the food-leaving assay [3,65], and the lipid-depleted *fat-6(lf);fat-7(lf)* males exhibited a behavioral preference for feeding over mating [61]. Unlike our phenotype, starving wild-type males and young adult *fat-6(lf);fat-7(lf)* males were fertile [61,66], suggesting the *ogt-1* fertility phenotype is more complex than starvation alone. Future studies could assess the interactions of *ogt-1* and nutrient sensing pathways such as insulin/IGF signaling, AMPK, and mTOR in the context of *C. elegans* male fertility.

Beyond our behavioral findings, additional processes may contribute to the reduction in fertility. We found that prior to mating, young adult *ogt-1* males had a sperm count significantly lower than wild-type (Fig 2D). Still, *ogt-1* males had an average sperm count over one thousand, which suggests this is unlikely to fully explain the low progeny counts we measured

for overnight mating (Fig 1A). In fact, this reduction is of similar magnitude to the reduction in hermaphrodite brood size that has been reported for the *ogt-1(ok430)* strain [24], raising the possibility that reduced sperm count is a phenotype common to males and hermaphrodites. It remains possible *ogt-1* sperm are defective in some way, as we did not directly test their activation, guidance, motility, or ability to fertilize oocytes. However, a sperm transfer assay suggested *ogt-1* male sperm are capable of properly localizing to the spermatheca (S4 Fig). We also assessed the male tail and found that deletion males looked normal with only a mild, allele-specific effect on rays (S3 Fig), though this does not rule out the possibility of subtle anatomical changes we were unable to identify.

The male fertility phenotype we have examined does not depend upon the catalytic activity of OGT-1 (Fig 4). OGT serves many roles within the cell, and increasing attention has been focused on the importance of its functions beyond GlcNAc transfer [14]. In vertebrates, OGT is required for proteolytic cleavage of HCF-1 [13], though this activity is absent in invertebrates such as *C. elegans* [67,68]. A recent study showed that beyond its enzymatic activities, OGT serves a role necessary for cell proliferation [14]. In addition to the catalytic domain, OGT consists of a TPR domain critical to substrate selection and is the site of protein-protein interactions, several post-translational modifications, and dimerization [8,69]. Interestingly, the majority of patient alleles associated with the congenital disorder OGT-XLID occur in the TPR domain [17], and it is unknown how non-catalytic roles could contribute to the disease. Because *C. elegans* is distinct among model organisms in tolerating loss of *O*-GlcNAcylation, it is uniquely suited to investigate non-catalytic roles of OGT. Thus, studies in *C. elegans* can contribute to our understanding of OGT-XLID disease etiology. In fact, prior studies have described non-catalytic roles of OGT including aldicarb sensitivity [30], adaptation to hypertonic stress [25], and entry into adult reproductive diapause [31]. Here we have shown that OGT-1 catalysis is dispensable for normal male fertility and mating behaviors including food-leaving, mate-response, and turning (Figs 4 and S8). The contributions of OGT-1 to *C. elegans* male fertility may include protein-protein interactions or gene-regulatory roles independent of *O*-GlcNAcylation. In human cells, several regulatory interactions have been documented between OGT and other proteins that are likely wholly (Ataxin-10 [70]) or partially independent (p120-catenin [71], mSin3A [72]) of OGT catalytic activity. Our results suggest there are important non-catalytic roles of OGT still to be discovered, which could be further investigated through targeted epistasis experiments and unbiased transcriptomics focused on males.

OGT's function as a nutrient sensor is multifaceted, and is often described in relation to UDP-GlcNAc concentration, which varies with cellular nutrient levels. Differing UDP-GlcNAc concentrations lead to OGT targeting different subsets of proteins for *O*-GlcNAc transfer [73]. This suggests more generally that OGT interacts with different proteins in high and low nutrient states, which could have implications relevant to our findings. Interestingly, the structurally similar TPR protein SSN6 is a component of a glucose-responsive repressor complex in yeast, and has been shown to have different binding partners depending on nutrient conditions [74]. In addition, expression of OGT is responsive to glucose and UDP-GlcNAc levels [73,75,76].

In conclusion, we have uncovered a previously uncharacterized defect in *C. elegans* male mating caused by *ogt-1* deletion. The phenotype was rescued by hypodermis-specific expression of an *ogt-1* transgene, demonstrating an additional contribution of the hypodermis to fertility. Our experiments demonstrate that catalytic activity of OGT-1 was not required for fertility, contributing to a better understanding of non-catalytic roles of the enzyme. Further exploration of these functions will uncover new pathways regulated by OGT and build towards a better understanding of how disruption of OGT leads to disease.

## Materials and methods

### Worm strains and maintenance

All worms were maintained at 20°C on NGM plates seeded with OP50 *E. coli* [77] unless otherwise noted. Strains were acquired through the Caenorhabditis Genetics Center at the University of Minnesota. Most strains were crossed into the *him-5(e1490)* background before they were used in experiments. Males from the following strains were used in this study: Bristol N2, CB1490 *him-5 (e1490) V*, DDK14 *ogt-1(jah01) III; him-5(e1490) V*, DDK11 *ogt-1(ok430) III; him-5(e1490) V*, DDK12 *ogt-1(ok1474) III; him-5(e1490) V*, DDK15 *him-5(e1490) V; oga-1 (av82) X*, DDK13 *him-5(e1490) V; oga-1(ok1207) X*, DDK18 *ogt-1(jah01) III; him-5(e1490) V; oga-1(ok1207) X*. Hermaphrodites from two strains, BA17 *fem-1(hc17) IV*, and PD3165 *unc-39 (ct73) V* were used in mating assays. The temperature-sensitive BA17 strain was maintained at the permissive temperature of 15°C [38,78]. For rescue experiments, we established the following transgenic lines carrying extrachromosomal arrays: DDK16 *ogt-1(jah01) III; him-5(e1490) V; jahEx1[ogt-1(fosmid) + myo-2p::gfp]*, DDK19 *ogt-1(jah01) III; him-5(e1490) V; jahEx2[ogt-1 (fosmid) + myo-2p::gfp]*, DDK23 *ogt-1(jah01) III; him-5(e1490) V; jahEx6[eft-3p::ogt-1 + eft-3p::gfp]*, DDK24 *ogt-1(jah01) III; him-5(e1490) V; jahEx7[eft-3p::ogt-1 + eft-3p::gfp]*, DDK25 *ogt-1(jah01) III; him-5(e1490) V; jahEx8[ges-1p::ogt-1 + ges-1p::gfp + myo-2p::gfp]*, DDK32 *ogt-1(jah01) III; him-5(e1490) V; jahEx11[dpy-7p::ogt-1 + dpy-7p::gfp + myo-2p:: mCherry]*, DDK33 *ogt-1(jah01) III; him-5(e1490) V; jahEx12[rgef-1p::ogt-1 + rgef-1p::gfp + myo-2p::mCherry]*, DDK27 *ogt-1(jah01) III; him-5(e1490) V; jahEx9[rgef-1p::ogt-1 + rgef-1p:: gfp + myo-2p::gfp]*, DDK31 *ogt-1(jah01) III; him-5(e1490) V; jahEx10[unc-119p::ogt-1 + unc-119p::gfp + myo-2p::mCherry]*, DDK52 *ogt-1(jah01) III; him-5(e1490) V; jahEx13[hlh-3p::ogt-1 + hlh-3p::gfp + myo-3p::mCherry]*, DDK53 *ogt-1(jah01) III; him-5(e1490) V; jahEx14[hlh-3p:: ogt-1 + hlh-3p::gfp + myo-3p::mCherry]*. The *ogt-1::gfp* fusion and *ogt-1* point mutant lines were kindly gifted by the Lamitina lab, then crossed to *him-5(e1490)* to establish several strains used in this study: DDK26 *ogt-1(dr84)[ogt-1::GFP] III; him-5(e1490) V*, DDK28 *ogt-1(dr84 [ogt-1::GFP] dr91[H612A]) III; him-5(e1490) V*, DDK29 *ogt-1(dr84[ogt-1::GFP] dr89[K957M]) III; him-5(e1490) V*, DDK40 *ogt-1(dr84[ogt-1::GFP] dr91[H612A]) III; him-5(e1490) V; oga-1 (av82) X*, DDK39 *ogt-1(dr84[ogt-1::GFP] dr89[K957M]) III; him-5(e1490) V; oga-1(av82) X*. PCR was used for genotyping most strains; primers are listed in S8 Table.

### *fem-1* male mating assay

To test the ability of male worms to sire progeny, they were allowed to mate with *fem-1 (hc17)* adults raised at 25°C which are effectively females, as they do not produce self-sperm [38,78,79]. For each cross, 15 males were plated with 5 *fem-1* animals on a 35mm plate with a small lawn of OP50 and allowed to mate overnight (18h) at 20°C. After mating, each *fem-1* worm was transferred to a new plate and allowed to lay eggs over 24h, at which time the worm was removed. Progeny were subsequently counted 24-48h later. Males used in these assays were approximately 1-day adults. For strains carrying extrachromosomal arrays, only males positive for the fluorescent marker were used. One-way ANOVA was used to statistically compare progeny counts from each male genotype to wild type (*him-5*), or to *ogt-1*.

### *ogt-1* rescue experiments

A 30kb genomic fragment fosmid (WRM0635dF05) containing wild-type *ogt-1* was injected into *ogt-1(jah01);him-5* hermaphrodites along with a fluorescent marker (*myo-2p::gfp*) to create two independent rescue lines.

In order to test tissue-specific expression of *ogt-1* on the fertility phenotype, several plasmids with different promoters were constructed. To insert the *ogt-1* gene into a plasmid, the longer isoform of the gene (K04G7.3a.1) and its 3'UTR (614bp) were amplified from cDNA. Primers added an XmaI restriction site before the start codon, and a SpeI restriction site following a putative polyadenylation site downstream of the 3'UTR. This PCR product was cloned into a pBlueScript plasmid backbone. As introns have been reported to enhance transgene expression [80], the endogenous introns 5 and 6 (each approximately 50bp) were inserted into the gene by amplifying a portion of the gene from genomic DNA and using restriction sites for StyI and BlpI within exonic sequence. This intron-containing plasmid was used for subsequent cloning, in which various promoter fragments were amplified from genomic DNA using primers to add a SalI site upstream and a XmaI site downstream. These same promoter fragments (with one added nucleotide to keep GFP in frame) were also cloned into the promoterless GFP vector pPD95.67 (originally provided by Dr. Andrew Fire) to co-inject with the *ogt-1* rescue plasmids. See S8 Table for the primers used for genotyping and cloning.

Lines with tissue-specific *ogt-1* expression were established by microinjection of these plasmids into *ogt-1(jah01);him-5* hermaphrodites. Each injection mix contained the *ogt-1* rescue plasmid, the GFP expression plasmid with the same promoter, a fluorescent marker such as *myo-2*::*mCherry*, and empty pBlueScript to dilute.

## Western blot

Samples used for western blotting were either approximately 200 picked adults per lane (Figs 1C and S1), or washed off one 6cm plate with many adults (Figs 4B and S7). Samples were then directly boiled in SDS loading dye (Quality Biological) w/ freshly added 2-mercaptoethanol to 2%. Samples were run on polyacrylamide gel in MOPS, then transferred to nitrocellulose at 100V for 100 minutes (Figs 1C and S1), or dry transferred using the iBlot 2 system (Figs 4B and S7). Primary antibodies targeting actin (Abcam ab8227) and *O*-GlcNAc (RL2, Thermo MA1-072) were used, followed by Odyssey secondary antibodies. The stained membrane was imaged by Odyssey.

## Immunohistochemistry

Worms were put in M9 buffer on poly-lysine treated slides and dissected using two needles to remove the head. Immediately following dissection, a coverslip was applied, and the slide was immersed in liquid nitrogen. The coverslip was then flicked off, and the slides were immersed in methanol for fixation. The RL2 antibody (Thermo MA1-072) was used to target *O*-GlcNAc, and tubulin was visualized with an antibody targeting TBB-2 kindly gifted by Dr. Kevin O'Connell. Fluorescent AlexaFluor secondary antibodies (Invitrogen) were used for visualization. Slides were mounted using Vectashield with DAPI (Vector labs) and imaged with a 20X objective on a Zeiss LSM 700 confocal microscope.

## Live worm imaging of male tail

On glass microscope slides, 2% or 5% agarose was used to make pads. Young adult males were picked into M9 containing anesthetic (0.1% tricaine and 0.01% tetramisole) on the agar pads, then a coverslip was applied, and the worms were immediately imaged with a 63X oil objective on a Zeiss LSM 700. Male tails were scored as deviant if any of the V6 rays were missing or fused.

## Sperm competition

Since hermaphrodites produce their own sperm, *C. elegans* male sperm have a competitive advantage to preferentially fertilize oocytes [81]. To assess the competitiveness of *ogt-1* male sperm [82], males were crossed with self-fertile hermaphrodites with a scorable phenotype: the uncoordinated phenotype (Unc) of *unc-39(ct73)* homozygotes. Crosses were performed overnight at 20˚C with 15 males and 5 *unc-39(ct73)* hermaphrodites on a 35mm plate with a small OP50 lawn. The 24h brood of each hermaphrodite was then allowed 24-72h to develop to make the Unc phenotype easier to discern. The number of Unc and non-Unc progeny was counted and reported as the percent of non-Unc progeny out of total progeny and compared by t-test. As *unc-39(ct73)* hermaphrodites used in this assay had relatively high rates of death, the data reported represents only those that were still alive and healthy after the 24h period of egg laying.

## Sperm transfer

As has been previously described [83], MitoTracker was used to stain live males and thus fluorescently label their sperm. Males were stained with 10μM MitoTracker Red (Invitrogen) in a watch glass for two hours, then put on a normal NGM plate to recover overnight. Twenty-four males were then plated with six unlabeled mates for one hour. We chose to use *fem-1* females for consistency with other experiments and to reduce the chance that ovulation would displace sperm. *fem-1* worms were then separated from the males and allowed to recover for one hour before mounting on 2% agar pads and imaged with a 20X objective on a Zeiss LSM 700. For this experiment, sperm transfer success was defined as any MitoTracker-labeled sperm present in the reproductive tract of the *fem-1* animal. The number of instances of mating success and failure by each male genotype were compared using Fisher's exact test. Quantification of fluorescence signal was done using ImageJ to determine mean fluorescence intensity of a defined region and corrected for background.

## DAPI staining of virgin males for sperm count

To ensure males are virgin prior to assessment of sperm count, synchronized males at the L4 stage were picked onto a plate and allowed to develop into adults overnight. These adult virgin males were then dehydrated by applying successively higher ethanol concentrations, then fixed with acetone and stained with 20μg/μL DAPI in PBS. Each slide was imaged with a 63X oil objective on a Zeiss LSM 700 as a z-stack with 0.2um steps. Two methods were used for analysis of these images: either manual counting using the multiselect tool in ImageJ, or with Imaris software using a published method for counting germline nuclei [84].

## Mating behavior

The food-leaving assay (also known as simply the leaving assay) [3] was used to score mate-searching behavior. 9cm plates were poured thin with 10mL of NGM, seeded the next day with 2μL of OP50 liquid culture, then used in the assay the following day. A single male was placed in the approximately 4mm OP50 lawn and monitored over 24h to see if it had engaged in food-leaving behavior by looking at the tracks left by the male on the plate by eye. Plates were checked every hour for the first 8 hours, then at the 12h and 24h timepoints. A male is considered a "leaver" if its tracks reach 1cm or closer to the edge of the plate at some point over the course of the assay. The number of leavers and non-leavers for each male genotype at the endpoint were compared using Fisher's exact test. Curve comparison between multiple genotypes was performed with the log-rank Mantel-Cox test.

Mating initiation was scored by the mate-response assay (also known in the literature as the latency assay or the response to contact assay) [53,54]. Ten *fem-1* worms were plated on a 35mm NGM plate with a small OP50 lawn and allowed an hour to condition the plate prior to the start of the assay. For each assay, 5 males (either WT or *ogt-1*) were added to the center of the OP50 lawn and allowed up to 6 minutes to initiate mating, here defined as 2 seconds of contact between the ventral side of the male tail and the mate. Males that initiated mating were removed from the plate during the assay. The proportion of mating initiation and failure to initiate between genotypes was compared using Fisher's exact test.

The average movement speed of individual males was determined by recording and analyzing videos of adult males moving freely on an unseeded NGM plate. To capture video, an Olympus DP74 camera was used with an Olympus SZX16 dissecting microscope. The videos were analyzed using the wrMtrck software package for ImageJ [52], which tracked the movement of each individual male throughout the duration of each video.

Video analysis was used to assess turns, location of vulva (Lov), and time spent at the vulva. Videos were recorded with the Olympus DP74 camera on an Olympus SZX16 dissecting microscope over five- or ten-minute intervals. Several *fem-1* worms were anesthetized with 0.1% tricaine and 0.01% tetramisole just prior to the experiment and were distributed evenly on the OP50 lawn throughout the viewing area. An equal number of males were then added to the plate and the recording started. Videos taken on the same day alternated between genotypes, using the same plate and *fem-1* animals for consistency. Initiation was defined as contact between the male tail and its mate and the start of backward locomotion. Turns were assessed as "good", "sloppy", or "missed" as has been defined in prior studies of turning behavior [55]. The total number of turns in each category for each genotype was compared by Chi-square analysis. The ability of a male to locate the vulva (location of vulva or LOV) was scored by the ratio between the number of times the male stopped at the vulva to the total number of times the tail passed over the vulva without stopping [50,85], and compared with Fisher's exact test. Dwell time at the vulva was defined as the interval from when the male located the vulva to the time the male disengaged from the vulva, and compared by t-test. In cases where the video ended before the male disconnects, or where another male knocks the male off, these dwell time values were excluded from the analysis. We also report the total number of behaviors (initiations, turns, vulval locations) per genotype, normalized to the total number of males analyzed in the videos (S5 Table).

## Statistical analysis

Comparisons between groups were performed with t-test for two groups, or one-way ANOVA for three or more groups, using GraphPad Prism 9 (GraphPad Software, Inc., La Jolla, CA). When comparing binary categorical data, Fischer's exact test was used. Chi-square was used for turn quality analysis (S6A and S8B Figs), which included three categories. For Figs 4D and S5, curve comparison was performed with the log-rank Mantel-Cox test.

## Supporting information

**S1 Fig. Uncropped western blot with fosmid rescue line from Fig 1C.** Left: O-GlcNAc (RL2 antibody), right: actin antibody. All genotypes have *him-5* in background.
(TIF)

**S2 Fig. Immunohistochemistry of *ogt-1(fosmid)* line, with all channels shown.** Representative immunohistochemistry images of dissected males, as shown in Fig 1D, including O-GlcNAc (RL2 antibody, in green), tubulin (anti-TBB-2 antibody, in red), DAPI (in blue),

and all three fluorescent channels merged with differential interference contrast (DIC, grey-scale). All worms are in the *him-5* background.
(TIF)

**S3 Fig. One *ogt-1* allele showed increased incidence of ray development defects.** (A) Percentage of males with normal V6 ray morphology, with total number of observed normal and total males of each genotype. Defects include ray fusions and missing rays. Ratio of normal to deviant males was compared to wild-type with Fisher's exact test and is shown above. ** = p<0.01, all others not significant. (B) Representative differential interference contrast (DIC) image of a normal male tail for wild-type control and a severe example of fused rays (arrow) in the *ogt-1(ok1474)* line. All males are in the *him-5* background.
(TIF)

**S4 Fig. *ogt-1* males transfer fewer sperm, sperm localizes to spermatheca.** Three representative differential interference contrast DIC (greyscale) images of *fem-1* animals after successful sperm transfer from MitoTracker-labelled (magenta) males of indicated genotypes. Yellow dashed lines indicate spermathecae. These images are examples of successful sperm transfer, excluding those with no sperm in their reproductive tract. Percent success rate (as shown in Fig 2C) is included with genotype above images. Right, quantification of MitoTracker signal in the reproductive tract of *fem-1* worms after 1h mating, including only data from images of successful sperm transfer.
(TIF)

**S5 Fig. *ogt-1* males are less likely to leave food.** Time course of the food-leaving assay, of which the 24h timepoint is shown in Fig 3B. Each point represents the percent remaining at each timepoint, with data summed from six replications of the food-leaving assay. Three replications of the experiment are not included in this figure, as they did not include all intermediate time points (all nine experiments are included in Fig 3B). Data was collected for each hour from 0–8, and at 24h. Sample size, final percent leavers, and statistical comparison are shown to the right. All subjects are males in the *him-5* background. Curve comparison between genotypes performed with log-rank Mantel-Cox test. **** = p<0.0001.
(TIF)

**S6 Fig. *ogt-1* males showed aberrant turning behavior and normal location of vulva.** (A) *ogt-1(jah01)* males have a higher incidence of sloppy turns (loss of contact, but successful turn) and of missed turns (complete loss of contact with mate), as assessed by Chi-square. (B) *ogt-1 (jah01)* males have normal vulval location efficiency, shown as the percent of vulva encounters in which the male showed successful location of vulva (LOV) versus the percent in which the male does not stop at the vulva (pass), as assessed by Fisher's exact test. (C) *ogt-1(jah01)* and wild-type males dwell at the vulva for similar lengths of time, as assessed by t-test. Dwell time is defined as the time from when the male stops at the vulva to the time the male leaves the vulva. All males are in the *him-5* background. **** = p<0.0001, ns = not significant.
(TIF)

**S7 Fig. Uncropped western blot with *ogt-1* point mutant lines from Fig 4B.** Left: O-GlcNAc (RL2 antibody), right: actin antibody. Genotypes of strains given above with regard to the *ogt-1* and *oga-1* genes. gfp = *ogt-1(dr84)*, Δ = deletion (*ogt-1(jah01)* or *oga-1(av82)*), H612A = *ogt-1(dr91)*, K957M = *ogt-1(dr89)*. All strains are in the *him-5* background.
(TIF)

**S8 Fig. Additional behavioral tests of catalytic-dead and hypodermal-rescue *ogt-1* lines.** (A) Repeat of the mate-response assay, including *ogt-1(K957M)* and *dpy-7p*::*ogt-1*, compared

with wild-type and with *ogt-1(jah01)* by pairwise Fisher's exact test. Results include five independent replications of the experiment. (B) Scoring of turn quality in videos of males mating with anesthetized *fem-1* animals, shown as percent of total turns assessed for each genotype. Good = successful turn with no loss of contact, sloppy = successful turn with loss of contact, missed = failed turn where male loses contact with mate. The experimenter was blinded to the genotype of the males while scoring turns. Statistical comparisons to WT and *ogt-1* were performed with pairwise Chi-square analyses. * = p<0.05, ** = p<0.01, *** = p<0.001, ns = not significant. P-values between 0.05 and 0.10 are shown on the graph. *ogt-1(K957M) = ogt-1 (dr89)*. All strains are in the *him-5* background, and *dpy-7p::ogt-1* is additionally in the *ogt-1* background.
(TIF)

**S1 Table. Data underlying Fig 1.**
(XLSX)

**S2 Table. Data underlying Fig 2.**
(XLSX)

**S3 Table. Data underlying Figs 3 and S5.**
(XLSX)

**S4 Table. Data underlying Fig 4.**
(XLSX)

**S5 Table. Data from analysis of mating videos, data underlying S6 Fig.**
(XLSX)

**S6 Table. Data underlying S8A Fig.**
(XLSX)

**S7 Table. Data underlying S8B Fig.**
(XLSX)

**S8 Table. DNA oligos used in this study.**
(XLSX)

## Acknowledgments

We thank the members of the Hanover lab, Dr. Andy Golden, Dr. Xiaofei Bai, and the other members of the Golden lab, Dr. Harold Smith (NIDDK Genomics Core), and the Baltimore Worm Club for helpful discussions, technical assistance, reagents, and strains. We thank Dr. Kevin O'Connell for his help with immunohistochemistry and for sharing his anti-TBB-2 antibody. We thank the Lamitina lab for sharing strains.

## Author Contributions

**Conceptualization:** Daniel Konzman, Tetsunari Fukushige, Michael Krause, John A. Hanover.

**Data curation:** Daniel Konzman.

**Formal analysis:** Daniel Konzman, Mesgana Dagnachew.

**Investigation:** Daniel Konzman.

**Methodology:** Daniel Konzman, Tetsunari Fukushige, Michael Krause.

**Project administration:** John A. Hanover.

**Resources:** John A. Hanover.

**Supervision:** Tetsunari Fukushige, Michael Krause, John A. Hanover.

**Visualization:** Daniel Konzman.

**Writing – original draft:** Daniel Konzman.

**Writing – review & editing:** Daniel Konzman, Tetsunari Fukushige, Michael Krause, John A. Hanover.

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
