## [Decision Letter · Decision Letter 0]

14 Jun 2022

Dear Dr %Hanover%,

Thank you very much for submitting your Research Article entitled 'O-GlcNAc transferase plays a non-catalytic role in C. elegans male fertility' to PLOS Genetics.

The manuscript was fully evaluated at the editorial level and by independent peer reviewers. The reviewers appreciated the attention to an important problem, but raised some substantial concerns about the current manuscript. Based on the reviews, we will not be able to accept this version of the manuscript, but we would be willing to review a much-revised version. We cannot, of course, promise publication at that time.

Should you decide to revise the manuscript for further consideration here, your revisions should address the specific points made by each reviewer.  Among the points raised by the reviewers, two key issues stand out as minimal revisions that are needed for re-considering the manuscript: 1) providing more definitive results for fuction of ogt-1 in the hypodermis for the noted male mating phenotypes, and 2) better characterization of the mating defects (issues raised by both Reviewers #2 and #3). We will also require a detailed list of your responses to the review comments and a description of the changes you have made in the manuscript.

If you decide to revise the manuscript for further consideration at PLOS Genetics, please aim to resubmit within the next 60 days, unless it will take extra time to address the concerns of the reviewers, in which case we would appreciate an expected resubmission date by email to plosgenetics@plos.org.

[LINK]

We are sorry that we cannot be more positive about your manuscript at this stage. Please do not hesitate to contact us if you have any concerns or questions.

Yours sincerely,

Kaveh Ashrafi

Associate Editor

PLOS Genetics

Gregory P. Copenhaver

Editor-in-Chief

PLOS Genetics

Reviewer's Responses to Questions

**Comments to the Authors:**

Reviewer #1: The manuscript “O-GlcNAc transferase plays a non-catalytic role in C. elegans male fertility” describes the reproductive defects of C. elegans males that lack the gene ogt-1, which encodes the enzyme O-GlcNAc transferase. From behavioral analyses of progeny counts, dwell times of solitary males in bacterial lawns, and visual inspection of copulation behavior, the authors found that ogt-1 deletion males have a reduced behavioral drive to initiate and sustain some of the earlier sensory-motor steps of mating. The male mating deficiencies can be partially ameliorated by copulations with mobility impaired mates, but with mobile females/hermaphrodites, the behavioral defects render many ogt-1 mutant males near infertile. The authors found that the site-of-action for OGT-1 is the male hypodermis/epidermis and interestingly, the utility of the molecule for mating behavior does not rely on its catalytic ability to transfer acetylglucosamine to other molecules. The behavioral assays are conducted and analyzed competently, and the finding that O-GlcNAc transferase-deficient OGT-1 protein can still promote male mating behavior is unexpected, exciting and implies that the molecule has additional metabolic/regulatory functions that can be further determined by studying male reproductive behavior.

I have a few very small issues that can be address via writing.

Page 34. Line 939. “… is placed in a small dot of OP50 (2ul) in the center…” Best describe in parentheses the small dot as a diameter of the lawn rather than a volume of inoculum.

For Figure 2 C, Need to clarify if the sperm transfer efficiency assay conducted similarly as the progeny count assay? Was this still the same 15 males to 5 female ratio for 18 hours?

Page 11 line 246. The statement “ These findings suggest the later steps proceed normally in ogt-1 males, supporting…” might need to be soften a bit. The only way to make that statement confidently is to do single male/single female mating analysis in order to determine that any exceptional ogt-1(null) male that successfully intromit its spicules and transfer sperm sires a statically comparable brood relative to a single wild-type male. Most of the assays in the paper are done with couplings of 3:1 ratio of males to mates for 18 hours, so it is difficult to determine how many repeated matings (i.e with accumulated sperm) each female engaged in.

In the results or discussion, one needs to tell the reader what additional mutant phenotypes (altered lipids, altered glycogen, osmotic response differences… etc.) if any that ogt-1 catalytic dead mutants share with the ogt-1 deletion, other than the loss of sugar modification. Is mating behavior the only thing not disrupted by the catalytic point mutations? This would be important since it would allow the reader to correlate any secondary phenotypes that might impinge with copulation behavior.

Page 19. Line 430. The statement “…suggesting the hypodermis plays a more important role in fertility than has been previously appreciated.” needs to be modified. Work has been published studying how nutrition and metabolism in the hypodermis/epidermis affects C. elegans male neural muscular circuitry involved with reproductive behavior and fertility. For example, similar to the ogt-1(null) mutants, day 1 fat-6(null), fat-7(null), (delta 9 fatty acid desaturase) mutant males show higher food preference to mating than wild type males (Figure 3 using a food/mate choice assay rather than the leaving assay, in Goncalves et al. 2022 iScience 25, 104082). Like OGT-1, the fatty acid desaturase site of action is the hypodermis/epidermis.

Additionally, phosphoenolpyruvate carboxykinase, a metabolic enzyme involved in glycerol/gluconeogenesis, required for lipid and sugar stores, has a site of action that is also in the male hypodermis/epidermis, but to sustain male copulation motor responses on day 2 of adulthood, rather than on day 1 (Goncalves et al 2020. iScience 23, 100990).

Signed L. Rene Garcia.

Reviewer #2: This work analyses the role of ogt-1, a O-GlcNAcyl transferase, in C. elegans male fertility. The experiments show that ogt-1 mutant males shire less progeny than wildtype ones and that this is due to behavioural defects rather than sperm quality. ogt-1 males display several mating defects associated with reduced mating drive (i.e. poor response to contact and low food-leaving to explore in search of mates). Through rescue experiments, they show that ogt-1 likely acts in the hypodermis. An interesting aspect of the work, in addition, is that the reduced fertility and food leaving behaviour are independent of enzymatic activity, suggesting other important roles for this protein beyond O-GlcNAcylation. These experiments are convincingly performed through CRISPR- generated mutants and biochemical analysis of their acetylation profiles. Overall, the manuscript is rigorous, the experiments are carefully performed and the conclusions are backed up by the data. There are only a couple of points that need revisiting.

- The tissue-specific rescues are performed with constructs that include ogt-1 introns that are potential regulatory regions of expression. Therefore, it is unclear whether the construct driving expression under the hypodermal promoter drives expression exclusively in hypodermis or also in other tissues (intron-driven expression). So, while the conclusion that the hypodermis is necessary, is correct, it is unclear whether this is sufficient. Could the authors try rescuing with a cDNA lacking all introns so to get rid of potential regulatory regions other than the hypodermis promoter?

- The catalytic dead mutants should be tested for the other phenotypes identified in the ogt-1 deletion mutants (i.e. response and turns) to better dissect the contribution of the different protein domains to the whole male phenotype.

- The authors show that ogt-1 males display a binomial distribution in their outcrossing success and this is due to differences in their ability to mate. Is the normalisation of response initiations per male counting all males an appropriate measure to perform and do statistical analysis, then? Since there are two different populations of males.

Reviewer #3: Konzman et al. report on the effects of the ogt-1 mutation on the reproductive performance in C. elegans males. The topic of this study is compelling because O-GlcNAc transferase is a nutrient sensor. However, I have concerns about interpretations of several experiments as well as the overall advance presented in this manuscript.

To start, dramatically reduced broods sired by ogt-1 males suggest that they have one or more substantial reproductive defects. For comparison, ogt-1 hermaphrodites suffer ~25-30% reduction in brood size, so the effect of the mutation appears to be considerably stronger in the males. To narrow down the site of ogt-1 action wrt reproductive performance in males, the authors expressed ogt-1 under control of several tissue-restricted promoters. Expression in the hypodermis (dpy-7) appears to substantially rescue the phenotype. Yet, it is not clear what this result tells us because: 1) no obvious hypodermal defects are evident in ogt-1 males, 2) dpy-7 is expressed beyond the hypodermis, 3) expression in other cell types, notably the neurons, may offer at least some amelioration of the ogt-1 defect, particularly considering the variability of the test (strain #1 vs. strain #2 and individual to individual carrying the same transgene). I think a fair take on Fig. 1 is that the hypodermis plays some unclear role in the ogt-1’s reproductive function, while the contributions from other cell types haven’t been convincingly ruled out.

The authors next focus on the idea that behavioral defects are the cause of reduced reproductive performance of ogt-1 males. The conclusion in line 185 “initiation of mating was a significant barrier to reproduction for ogt-1 males” does not appear to be warranted by the results. Given the data presented up to this point, it is possible that the brood size defect of ogt-1 males is due to inability to sustain mating, rather than to initiate it.

The results described in lines 193-194 and presented in Fig. 2D do not rule out sperm defects. First, based on the description provided, we are left to assume that all germline nuclei were counted. Second, there appears to be a reasonable indication that ogt-1 males have fewer germline nuclei, and that the lack of significance is due to high interindividual variability and very low sample size. Third, even if ogt-1 males have approx. as many sperm as WT, those sperm could have substantial defects, for instance in movement or guidance.

The paragraph that starts on line 207 seemingly interchangeably uses terms “leaving assay”, “searching behavior”, and “exploratory behavior”. Since the equivalence between these terms has not been firmly established, it would be best to refer to the results of the assay as such.

The inference (line 215) that the somewhat higher % of leavers among ogt-1 males is due to “reduced mating drive” is not supported by the data presented in the preceding section, because this experiment did not test mating drive.

The inference (line 222) that the observation of fewer initiated matings by ogt-1 males is due to the fact that “ogt-1 males don’t seek mates as actively as wild-type” conflates results of the experiment described in the paragraph that starts on line 216 with the previous experiment. This conclusion does not appear to be warranted by the data.

I found the separation of Table 1 vs. Fig. 4A, 4C to be somewhat confusing. The authors do explain the difference between the data presented in the figure and the table, but perhaps there is a way to streamline this part of the presentation.

It seems that in Fig. 4A the two distributions are quite similar, but the top five values for WT shift the mean upward. Is this the case?

Were experiments in Fig. 4A and, particularly, Fig. 4B blinded?

The manuscript ends with an intriguing suggestion that the reproductive defects in males may be due to ogt-1 functions that do not depend on the catalytic domain, but it remains unclear what those functions may be.

My overall impression of the manuscript is that it highlights an interesting phenotype – a largely male-specific fecundity defect in ogt-1 mutants – that is caused by largely unclear mechanism(s). It could be due to gamete defect(s) as well as several behavioral defects acting in unknown cells. Given the broad expression of ogt-1, this remains perhaps the most plausible interpretation of the results presented in this manuscript. Overall, this study offers some, but modest insights into the mechanisms by which ogt-1 acts in male reproduction in C. elegans.

Minor comments:

Figures did not render well in the PDF version of the manuscript. This made it difficult to follow the finer points depicted in those figures.

The manuscript could benefit from additional editing. Two examples to illustrate this point - “the many different tissues” (Line 124) and “To determine which tissue ogt-1 expression is critical” (Line 124). There are several others.

At seven pages, the Discussion section is as long as the Results section.

Part of the manuscript’s motivation seems to be the use of C. elegans ogt-1 as a model for human O-GlcNAc transferase. It is not clear how well the phenotypes discussed in this manuscript advance this case. It may be better to focus on deeper understanding of the role(s) of ogt-1 in male fertility in C. elegans.

**Have all data underlying the figures and results presented in the manuscript been provided?**

Reviewer #1: Yes

Reviewer #2: Yes

Reviewer #3: Yes

PLOS authors have the option to publish the peer review history of their article (what does this mean?). If published, this will include your full peer review and any attached files.

Reviewer #1: **Yes: **L. Rene Garcia

Reviewer #2: No

Reviewer #3: No

---

## [Decision Letter · Decision Letter 1]

8 Nov 2022

Dear Dr %Hanover%,

We are pleased to inform you that your manuscript entitled "O- GlcNAc transferase plays a non-catalytic role in C. elegans male fertility" has been editorially accepted for publication in PLOS Genetics. Congratulations!

Yours sincerely,

Kaveh Ashrafi

Academic Editor

PLOS Genetics

Gregory P. Copenhaver

Editor-in-Chief

PLOS Genetics

Comments from the reviewers (if applicable):

Reviewer's Responses to Questions

**Comments to the Authors:**

Reviewer #1: The authors addressed all of my comments. Good job.

Reviewer #2: the authors have addressed my initial concerns as well as those of other reviewers and the manuscript is much improved. I have no further comments

**Have all data underlying the figures and results presented in the manuscript been provided?**

Reviewer #1: Yes

Reviewer #2: None

PLOS authors have the option to publish the peer review history of their article (what does this mean?). If published, this will include your full peer review and any attached files.

Reviewer #1: **Yes: **L. Rene Garcia

Reviewer #2: No

**Data Deposition**

http://datadryad.org/submit?journalID=pgenetics&manu=PGENETICS-D-22-00613R1

**Press Queries**

---

## [Editor Report · Acceptance letter]

11 Nov 2022

PGENETICS-D-22-00613R1 

O-GlcNAc transferase plays a non-catalytic role in C. elegans male fertility 

Dear Dr Hanover, 

We are pleased to inform you that your manuscript entitled "O-GlcNAc transferase plays a non-catalytic role in C. elegans male fertility" has been formally accepted for publication in PLOS Genetics! Your manuscript is now with our production department and you will be notified of the publication date in due course.

With kind regards,

Zsofi Zombor

PLOS Genetics

On behalf of:
